# Chronic Treatment with Melatonin Improves Hippocampal Neurogenesis in the Aged Brain and Under Neurodegeneration

**DOI:** 10.3390/molecules27175543

**Published:** 2022-08-29

**Authors:** Cristina Cachán-Vega, Ignacio Vega-Naredo, Yaiza Potes, Juan Carlos Bermejo-Millo, Adrian Rubio-González, Claudia García-González, Eduardo Antuña, Manuel Bermúdez, José Gutiérrez-Rodríguez, José Antonio Boga, Ana Coto-Montes, Beatriz Caballero

**Affiliations:** 1Department of Morphology and Cell Biology, University of Oviedo, 33006 Oviedo, Asturias, Spain; 2Instituto de Investigación Sanitaria del Principado de Asturias (ISPA), 33011 Oviedo, Asturias, Spain; 3Instituto de Neurociencias del Principado de Asturias (INEUROPA), 33006 Oviedo, Asturias, Spain; 4Hospital Monte Naranco, 33012 Oviedo, Asturias, Spain

**Keywords:** aging, adult hippocampal neurogenesis, neurodegeneration, melatonin

## Abstract

Adult hippocampal neurogenesis is altered during aging and under different neuropsychiatric and neurodegenerative diseases. Melatonin shows neurogenic and neuroprotective properties during aging and neuropathological conditions. In this study, we evaluated the effects of chronic treatment with melatonin on different markers of neurodegeneration and hippocampal neurogenesis using immunohistochemistry in the aged and neurodegenerative brains of SAMP8 mice, which is an animal model of accelerated senescence that mimics aging-related Alzheimer’s pathology. Neurodegenerative processes observed in the brains of aged SAMP8 mice at 10 months of age include the presence of damaged neurons, disorganization in the layers of the brain cortex, alterations in neural processes and the length of neuronal prolongations and β-amyloid accumulation in the cortex and hippocampus. This neurodegeneration may be associated with neurogenic responses in the hippocampal dentate gyrus of these mice, since we observed a neurogenic niche of neural stem and progenitor/precursors cells in the hippocampus of SAMP8 mice. However, hippocampal neurogenesis seems to be compromised due to alterations in the cell survival, migration and/or neuronal maturation of neural precursor cells due to the neurodegeneration levels in these mice. Chronic treatment with melatonin for 9 months decreased these neurodegenerative processes and the neurodegeneration-induced neurogenic response. Noticeably, melatonin also induced recovery in the functionality of adult hippocampal neurogenesis in aged SAMP8 mice.

## 1. Introduction

Adult neurogenesis was not widely accepted until the late 1990s [1]. Since then, numerous studies have provided substantial evidence supporting the presence of neural stem/progenitor cells in the mammalian adult brain, including rodent, primate, and human brains [1,2,3,4,5,6]. The current dogma “that new neurons can and do form in the adult mammalian brain” [2] has increased interest in research on adult neurogenesis. Adult neurogenesis is a process that starts with cell proliferation and ends with new functional neurons that integrate into existing neural circuits. There are two “canonical” regions of the mammalian adult brain that generate new neurons: (a) the border of the lateral ventricles of the brain (subventricular zone) and (b) the subgranular zone of the hippocampal dentate gyrus [1,3,4,7]. Several non-canonical regions also contain neural progenitor cells, including the neocortex, striatum, and hypothalamus [1].

In the adult hippocampal neurogenesis, the differentiation of neural stem cells into mature functional neurons occurs via a clearly defined set of cellular stages from Type-1 cells (radial glia-like cells), to Type-2a and 2b cells (neural progenitor cells), to Type-3 cells (neuroblasts), which undergo migration and final maturation to functional neurons [1,3,7]. This set of cellular states can be differentiated by the sequential activation/inactivation of the expression of different molecular markers, including Glial Fibrillar Acidic Protein (GFAP), Nestin, T-Box transcriptional Factor 2 (TBR-2), Doublecortin (DCX), Neurogenic Differentiation factor 1 (NeuroD1), β-Tubulin III, Neuronal Nuclei (NeuN), Calbindin, and Calretinin, among several others [1,3,7]. The new neurons developing from neural stem cells in the subgranular zone will integrate into pre-existing neural networks of the granular neurons layer of the dentate gyrus in order to participate in learning and memory processes [1,4,6].

Although hippocampal neurogenesis persists in aged adults and Alzheimer’s disease (AD) patients [5], several impairments in the hippocampal neurogenic response have also been described during normal aging and under different neuropathological conditions [3,4,6,8,9]. Adult hippocampal neurogenesis is also impaired in the brains of patients with epilepsy, depression, ischemia, addictions, and cancer [6]. Neurogenesis alteration may be a consequence of a decrease in the pool of neural stem cells, alterations of the molecular micro-environment that do not favor cell proliferation and/or cell differentiation, or due to neural stem/progenitor cells that cannot respond to neurogenic signals in the aged or neurodegenerative brain [3]. Notably, AD is the most common form of dementia in elderly individuals, and the age-dependent decline in adult hippocampal neurogenesis may be further accelerated under this neurodegenerative disease, which contributes to hippocampus-dependent cognitive and emotional dysfunctions [5,6]. Therefore, interventions that promote an increase in the adult hippocampal neurogenesis in animal models, such as physical exercise or by stimulating learning processes, are key to improve hippocampal cognitive functions, even in the late phases of aging and especially under a situation of neurodegeneration [3,4,6]. In this regard, treatments with melatonin seem to be one of the most appropriate strategies, due to melatonin having relevant beneficial effects on neurogenesis impairments in several in vitro and in vivo studies in models of aging and different neurological disorders [8,10,11,12,13,14,15,16]. The role of melatonin in adult neurogenesis was confirmed in pinealectomized rats, which showed a decline in both melatonin levels as well as adult hippocampal neurogenesis. However, treatment with exogenous melatonin in these animals reversed the neurogenesis impairment [17]. In addition, melatonin has a broad neuroprotective effect against central nervous system disorders, particularly during aging and under different neurodegenerative conditions [8], due to its well-known antioxidant, anti-inflammatory, and anti-apoptotic properties [18,19,20,21]. Given these premises, we evaluated the effects of chronic treatment with melatonin on different markers of neurodegeneration and adult hippocampal neurogenesis in the aged and neurodegenerative brains of SAMP8 mice, which is an animal model of accelerated senescence that mimics aging-related late-onset AD pathology [22,23,24,25].

## 2. Results

### 2.1. Markers of Neurodegeneration in SAMP8 Mice

#### 2.1.1. β-Tubulin III Immunostaining in the Brain Cortex of SAMP8 Mice

β-Tubulin III is a microtubule-associated protein of the tubulin family found almost exclusively in the neuronal cytoskeletal [26]. β-Tubulin staining in SAMP8 mice treated with vehicle (control mice) showed an important disorganization of the layers in the brain cortex as a consequence of their accelerated senescence. In this way, only the layers of brain cortex I, III, and VI were certainly identified (Figure 1a). However, these brain alterations were reversed in mice treated chronically with melatonin for 9 months, and all of the layers of the brain cortex (I-VI) were easily observed (Figure 1b). At higher magnifications, we observed neurons under neurodegeneration (“dark neurons”) in the brain cortex of control mice. Dark neurons showed β-Tubulin III staining in both the nucleus and cytosol as a consequence of relevant damage to their neuronal cytoarchitecture (Figure 1c, asterisks). Neurons in the brain cortex of melatonin-treated mice showed β-Tubulin III staining primarily in the cytosol and appropriate staining of their neuronal prolongations (Figure 1d, arrows). At 1000× magnifications, the neurodegenerative damage in control mice was evident by the presence of dark neurons (asterisks) and poorly developed neuronal processes (arrows) compared to melatonin-treated mice (Figure 1e,f). However, significant statistical differences were not detected in the levels of β-Tubulin III between control mice and melatonin-treated mice (Figure 1g). Notably, the frequency of cortical cells positive for β-Tubulin III with evident neural processes were 4.1-fold higher in melatonin-treated mice compared to control mice (Figure 1h, *p* < 0.001). Likewise, we observed a higher length of neural prolongations in cortical cells positive for β-Tubulin III in melatonin-treated mice (251.86 (mean) ± 75.94 (SD)) compared to control mice (65.52 (mean) ± 14.67 (SD)) (Figure 1i, *p* < 0.001).

#### 2.1.2. β-Amyloid (1-42) Immunostaining in the Brain Cortex of SAMP8 Mice

SAMP8 mice are considered an animal model of AD [25]. We observed an important accumulation of β-Amyloid (1-42) peptides in cortical neurons in the brain of control mice (Figure 2a,c). However, β-Amyloid (1-42) accumulation decreased in melatonin-treated mice (Figure 2b,d). Notably, β-Amyloid (1-42) peptides accumulated intraneuronally (Figure 2c) and were always at higher levels in the cortex of control mice compared to melatonin-treated mice (Figure 2e, *p* < 0.01).

#### 2.1.3. β-Amyloid (1-42) Immunostaining in the Hippocampal Dentate Gyrus of SAMP8 Mice

The intraneuronal accumulation of β-Amyloid (1-42) peptides was also observed in the hippocampal dentate gyrus of control mice, specifically in the granule neurons layer and hilar neurons (Figure 3a,c). However, the levels of β-Amyloid (1-42) decreased in both types of neurons in the melatonin-treated mice (Figure 3b,d). We observed statistically significant differences in the levels of β-Amyloid (1-42) peptides between control mice and melatonin-treated mice (Figure 3e, *p* < 0.01).

### 2.2. Markers of Adult Hippocampal Neurogenesis in SAMP8 Mice

#### 2.2.1. Nestin Immunostaining in the Hippocampal Dentate Gyrus of SAMP8 Mice

Nestin is a cytoskeletal protein that conforms to type VI intermediate filaments and is considered a marker of neural stem/progenitor cells [7]. Nestin staining was primarily observed in the granule neurons layer in the hippocampal dentate gyrus of control mice (Figure 4a). However, the highest intensity of Nestin was observed in the subgranular zone of the dentate gyrus (Figure 4a). Nestin was also detected in hilar neurons in the hippocampus of control mice (Figure 4a, arrows). Nestin staining decreased in the hippocampal dentate gyrus of melatonin-treated mice (Figure 4b). We observed statistically significant differences in the levels of Nestin between control mice and melatonin-treated mice (Figure 4c, *p* < 0.001).

#### 2.2.2. TBR-2 Immunostaining in the Hippocampal Dentate Gyrus of SAMP8 Mice

TBR-2 is a transcription factor that plays a crucial role in the proliferation and differentiation of neural progenitor cells [7]. TBR-2 staining was observed in the granule neurons layer in the hippocampal dentate gyrus of control mice (Figure 4d). TBR-2 staining was more intense in the subgranular zone (Figure 4d) and was also detected in hilar neurons in the hippocampal dentate gyrus of control mice (Figure 4d, arrows). TBR-2 staining decreased in melatonin-treated mice (Figure 4e). We observed statistically significant differences in the levels of TBR-2 between control mice and melatonin-treated mice (Figure 4f, *p* < 0.01).

#### 2.2.3. NeuroD1 Immunostaining in the Hippocampal Dentate Gyrus of SAMP8 Mice

The neurogenic transcription factor NeuroD1 is a marker of neural precursor cells or neuroblasts with the capacity of migration [1,7]. NeuroD1 staining was intensively observed in the granule neurons layer in the hippocampal dentate gyrus of control mice (Figure 4g). Melatonin-treated mice showed much lower levels of NeuroD1 staining in their granule neurons (Figure 4h). We observed statistically significant differences in the levels of NeuroD1 between control mice and melatonin-treated mice (Figure 4i, *p* < 0.001).

#### 2.2.4. β-Tubulin III Immunostaining in the Hippocampal Dentate Gyrus of SAMP8 Mice

β-Tubulin III staining was observed at high intensity in hilar neurons in the hippocampal dentate gyrus of control mice (Figure 5a) and melatonin-treated mice (Figure 5b). However, β-Tubulin III staining in the neuronal prolongations was more evident in hilar neurons of melatonin-treated mice compared to control mice (Figure 5a,b, detail in small boxes). In this way, the frequency of hilar cells positive for β-Tubulin III with evident neural processes were 8.3-fold higher in melatonin-treated mice compared to control mice (Figure 5c, *p* < 0.001). Likewise, we observed a higher length of neural prolongations in hilar cells positive for β-Tubulin III in melatonin-treated mice (220.32 (mean) ± 57.61 (SD)) compared to control mice (75.40 (mean) ± 19.57 (SD)) (Figure 5d, *p* < 0.001).

Neuronal β-Tubulin III is a cytoskeletal protein expressed in postmitotic immature and mature neurons and, due to this, it is considered an early marker of newly created neurons [7]. At higher magnifications, we observed cells positive for β-Tubulin III staining in the subgranular zone of the hippocampal dentate gyrus in control mice (Figure 6a, arrows) and melatonin-treated mice (Figure 6b, arrows). We also observed some β-Tubulin III-positive cells in the deepest layers of the granule neurons layer in the hippocampus of control mice (Figure 6c, arrow). However, there were a higher number of cells positive for β-Tubulin III in the subgranular zone of melatonin-treated mice compared to control mice (Figure 6b,d arrows; Figure 6e, *p* < 0.001). The total number of nuclei (nuclei volume) in the granule neurons layer (including the subgranular zone) was also significantly higher in melatonin-treated mice compared to control mice (Figure 6f, *p* < 0.05).

### 2.3. Principal Component Analysis

We applied the Principal Component Analysis (PCA) statistical dimension reduction tool to evaluate the associations between our study variables to correlate neurodegenerative and neurogenesis hippocampal markers (hippocampal β-Amyloid (1-42) peptides, Nestin, TBR-2, NeuroD1, and hippocampal β-Tubulin III levels). The PCA resulted in two main eigenvalues greater than 1 (Figure 7). These two components explained 75.42% of the total variance data (52.96% component 1 and 22.45% component 2).

As shown in Table 1, three variables (β-Amyloid (1-42), NeuroD1, and β-Tubulin III) loaded highest on the first component (component 1). The other two variables (TBR-2 and Nestin) loaded highest on the second component (component 2). The Kaiser–Meyer–Olkin (KMO) measure of sampling adequacy was 0.619, which showed a good and appropriate use of the factorial analysis with our sample data. Likewise, we observed a significance level of Bartlett’s test of sphericity (*p* < 0.001), which showed a significant and relevant correlation between the variables and our factor model (Table 1). Notably, our factor model showed a strong and significant correlation (0.707) between β-Amyloid (1-42) and NeuroD1 values (Table 2, *p* < 0.001). The β-Amyloid (1-42) levels also significantly correlated with Nestin and β-Tubulin III values (Table 2, *p* < 0.05). Nestin levels significantly correlated with TBR-2, NeuroD1 (Table 2, *p* < 0.01) and β-Tubulin III values (Table 2, *p* < 0.05). Finally, levels of NeuroD1 and β-Tubulin III were also significantly correlated (Table 2, *p* < 0.05). We also considered an additional PCA model by excluding values of β-Tubulin III in the dentate gyrus because the percentage of variance of this variable explained by our first PCA model was the lowest (45.1%). This second PCA model also separated our variables into the same two components, and increased the total variance explained by our factor model by 86.44% (data not shown), with good sampling adequacy (KMO, 0.550) and a significant and relevant correlation between the variables and the factor model (Bartlett’s test of sphericity, *p* < 0.001).

## 3. Discussion

SAMP8 mice are a good animal model for studying aging and age-related neurodegenerative processes [25]. Notably, SAMP8 mice exhibit many features that occur early in the pathogenesis of aging-related AD, such as oxidative stress, β-amyloid and α-synuclein accumulation, tau hyperphosphorylation, neurofibrillary tangles, gliosis, and cell death, and impairments in learning and memory [18,19,22,23,24,25]. In the present study, we planned to evaluate the response of adult hippocampal neurogenesis in this relevant animal model of neurodegeneration and effects of chronic melatonin treatment in adult hippocampal neurogenesis under neurodegeneration. In this way, we first corroborated neurodegenerative processes in the brains of SAMP8 mice. We studied neurodegeneration in the hippocampus (which is key for learning and memory) and in the brain cortex, because different zones of the cortex are implicated in several relevant brain functions such as movements, response to stimuli, and language and, thus, its neurodegeneration contributes to cognitive impairments in these animals. Our study found evident neurodegeneration in the brains of SAMP8 mice aged 10 months (control mice) based on the significant accumulation of β-amyloid (1-42) peptides in the cortex and hippocampal dentate gyrus. The biological hallmark of AD is the accumulation of β-amyloid peptides in specific brain zones, including the cortex and hippocampus [9]. In accord with our present data, Díaz-Moreno and collaborators (2013) demonstrated that β-amyloid peptides accumulated in the brains of SAMP8 mice, starting from 2–5 months of age [27]. Likewise, β-Tubulin III staining allowed us to observe other relevant neurodegenerative processes such as disorganization in the layers of the brain cortex, the presence of cortical dark neurons, and alterations in cortical and hippocampal neural processes and length of neuronal prolongations, as a consequence of impairments in the neuronal cytoskeleton. The cytoskeleton is the main intracellular structure that determines the morphology of neurons and maintains their integrity and, thus, disruption of its structure and function may underlie several neurodegenerative processes [28]. Therefore, we confirmed an age-related state of neurodegeneration in our animal model of ageing, as had already observed in previous studies [11,12,13,18,19,20,27,29,30,31,32,33,34,35,36,37].

Adult hippocampal neurogenesis decreases during normal aging as well as due to different neuropsychiatric and neurodegenerative conditions including AD, epilepsy, depression, ischemia, addictions, and even under stressful situations and sleep deprivation [3,4,6,8,9,38]. However, we have observed evident expression of several markers of the neurogenic response (Nestin, TBR-2, NeuroD1) in the granular neurons layer, especially in the subgranular zone, of the hippocampal dentate gyrus in control mice in our animal model of aging and neurodegeneration. Nestin is a well-known marker of neural stem and progenitor cells [7], while TBR-2 and NeuroD1 are transcription factors involved in neuronal lineage progression from multipotent stem cells [39]. Therefore, there appeared to be a varied niche of neural progenitor and precursor cells in the hippocampus of control mice, despite their accelerated senescence and neurodegeneration. Tobin and collaborators demonstrated that adult hippocampal neurogenesis persists in aged adults and AD patients [5]. Likewise, adult hippocampal neurogenesis can be stimulated by different physical and cognitive stimuli, even in the later phases of aging [3,4,40]. Notably, we observed a strong and significant positive correlation between β-amyloid (1-42) peptides and the levels of NeuroD1. Cells positive for NeuroD1 are considered neuroblasts that undergo marked morphological, electrophysiological, and gene expression changes that are associated with functional granule cells of the hippocampal dentate gyrus [1]. β-amyloid (1-42) peptides also correlated with Nestin and β-Tubulin III levels. Given these findings, our data suggest a neurogenic response in the hippocampus induced by the neurodegeneration observed in the aged brains of SAMP8 mice at 10 months of old. In accord with our data, the early accumulation of β-amyloid peptides in SAMP8 mice at 2 months of age can stimulate the proliferation of neural stem cells [27].

Our hypothesis that neurodegeneration activated neurogenic responses in the hippocampus of aged SAMP8 mice was firmly supported by the effect observed in chronically melatonin-treated SAMP8 mice, because these mice showed a decreased neurodegenerative level and, consequently, a lower neurogenic response compared to control mice. Our previous studies and several other authors had already demonstrated the neuroprotective benefits of long-time treatments with melatonin without side effects [11,12,13,18,19,20,29,30,31,32,33,34,35,36,37,38], and increases of both the half-life of SAMP8 mice (from 16 to 22 months) and their longevity (from 23 to 27 months) [38]. The present study showed that melatonin-treated aged mice had recovery in the layers of their brain cortex and decreased β-amyloid (1-42) accumulation in the cortex and hippocampal dentate gyrus. Likewise, β-Tubulin III showed improvement in neural processes and the length of neuronal prolongations in cortical and hilar neurons, which may improve the neural connectivity in the brain of melatonin-treated mice. In previous studies, melatonin also increased dendritogenesis in hilar neurons in hippocampal organotypic cultures [12,14]. These neuroprotective beneficial effects of our chronic treatment with melatonin in SAMP8 mice led to a decrease in the neurodegeneration-induced neurogenic response. This way, melatonin-treated mice showed a significant reduction in the expression of most of the neurogenic markers evaluated in the hippocampal dentate gyrus (Nestin, TBR-2 and NeuroD1).

However, melatonin improves neurogenesis in models of aging and neurodegenerative pathology during in vivo and in vitro experiments [8,38]. Notably, melatonin promoted cell viability, proliferation, and neuronal differentiation of neural stem cells in different studies in vitro [8,41]. Unlike our experimental conditions, all of these results were obtained under acute treatments of melatonin (from hours to 7 days) or using cultures of neural stem cells obtained from young animals (6–8 weeks). Therefore, acute treatments with melatonin in animal models of aging or neurodegenerative disease also promote a positive response in adult hippocampal neurogenesis [8,10,11,42]. Moreover, several studies by Ramirez-Rodriguez and collaborators also demonstrated a neurogenic effect of chronic treatments with melatonin (6–9 months) in adult BALB/c mice, which increased cell proliferation, survival, and maturation of newborn immature neurons. Melatonin also modulated the structural plasticity of the mossy fiber projection to establish functional synapses in the hippocampus of these mice [15,43,44]. Notably, in our present study the number of cells positive for β-Tubulin III in the subgranular zone was significantly increased by melatonin in the hippocampus of SAMP8 mice. β-Tubulin III is expressed in post-mitotic neurons even from an immature state [7]. Therefore, despite a reduced neurogenic response in the hippocampus of melatonin-treated mice, the process of hippocampal neurogenesis seems to be functional to produce neural progenitors (TBR-2-positive cells) and precursors (NeuroD1-positive cells) that survived to successfully mature toward immature neurons (β-Tubulin III-positive cells). Therefore, our chronic treatment with melatonin improved the functionality of adult hippocampal neurogenesis, as several other studies have shown [8,10,11,12,14,16,38,44].

The neurogenic effect of melatonin in the aged and neurodegenerative brains of SAMP8 mice may be supported by the promotion of cell survival and neuronal maturation of neural precursor cells. Our control mice, besides having relevant expression of Nestin, TBR-2, and NeuroD1 neurogenic markers, showed lower nuclei volume in the granular neurons layer (including the subgranular zone) and also a lower number of cells positive for β-Tubulin III in the subgranular zone compared to melatonin-treated mice. These results suggests that despite having a high number of neural stem cells and neural progenitor/precursor cells, the hippocampus of control SAMP8 mice has an alteration in cell survival and/or neuronal maturation process of neural precursor cells (NeuroD1-positive cells), which takes 2 to 8 weeks in mice [1]. Previous studies confirmed an alteration in hippocampal neurogenesis in SAMP8 mice with a preferential differentiation of neural stem cells into mature astrocytes, which contributed to their typical astrogliosis [45,46]. A block in the maturation of neuroblasts and reduced formation of immature neurons has also been described early in patients with AD that worsens with the progression of the disease [6]. Therefore, the survival and neuronal maturation processes of neural precursor cells may be compromised in the aged and neurodegenerative brains of SAMP8 control mice. Consistent with our present results, limited hippocampal neurogenesis in SAMP8 mice was previously observed starting from 5 months of age [45] and affected several processes of the neurogenic response, including survival of progenitor/precursor cells [47] and neuronal maturation of newly created immature neurons in the hippocampus [48]. Neurodegenerative processes driven by the accumulation of β-amyloid (1-42) peptides and phosphorylated Tau proteins in the brains of SAMP8 may also contribute to these alterations in adult hippocampal neurogenesis during aging [27,49].

Recent studies affirmed that mature neurons dedifferentiate (i.e., dematuration) to a pseudo-immature status and re-express the molecular markers of neural progenitor cells and immature neurons [50]. Interestingly, this process may occur in healthy individuals during aging and in the brains of patients with AD [50]. Our present data showed some Nestin-positive and TBR-2-positive cells in the hilus of the hippocampus of control SAMP8 mice, which may be a consequence of the dematuration of these mature neurons under neurodegeneration. Aberrations in the migration of new immature neurons into the hilus under neuropsychiatric conditions have also been described [51]. The lack of cells positive for β-Tubulin III in the subgranular zone of the dentate gyrus in the hippocampus of control SAMP8 mice may also be related to the aberrant migration of new created immature neurons to the hilus, instead of being properly integrated into the granular layer. The migration of immature neurons deeper into the granule neurons layer and their aberrant positioning in the hippocampus of SAMP8 mice was shown previously [46], as we also observed in the present study. These data support impairments in adult hippocampal neurogenesis in SAMP8 control mice at 10 months of age. Our chronic treatment with melatonin in these mice had a beneficial effect, decreased neurodegeneration, improved cell survival, and restored functional adult hippocampal neurogenesis by supporting the appropriate migration and neuronal maturation of neural precursor cells.

## 4. Materials and Methods

### 4.1. Animals

SAMP8 mice were obtained from the Council for SAM Research, Kyoto, Japan, through Harlan (Barcelona, Spain). The mice were housed in the Barcelona University facility under a 12/12-hr dark/light cycle, temperature-controlled (22 ± 1 °C), and were bred via brother–sister mating. Animals received tap water and a standard pellet diet ad libitum. Studies were performed by the Institutional Guidelines for the Care and Use of Laboratory Animals established by the European Communities Council Directive 2010/63/EU, Guidelines for the Care and Use of Mammals in Neuroscience and Behavioural Research, National Research Council 2003 and were also approved by the Animal Experimentation Ethics Committee (CEEA: 266/13) at the University of Barcelona and the Government of Catalunya (DAAM: 7149).

### 4.2. Treatment

Once newborn SAMP8 mice (*n* = 4 per treatment) were separated from their mothers (at 1 month of age), melatonin or vehicle treatments were initiated. Melatonin (Sigma, St Louis, MO, USA) was dissolved in a minimum volume of absolute ethanol in bottles protected from light and diluted in the drinking water to yield a dose of 10 mg/mL/kg during treatment from 1 to 10 months of age. SAMP8 mice drink about 5 mL/day and they weigh about 0.025 kg. Taking into account these data, the melatonin doses received was approximately 2 mg/day. The concentration of ethanol in the final solution was 0.066% (*v*/*v*). SAMP8 mice were decapitated at 10 months of age and the brains were immediately removed. The brains were fixed in 4% (*v*/*v*) paraformaldehyde in phosphate buffer (PBS) pH 7.4 for at least 24 h and washed in PBS buffer. The samples were embedded in paraffin using standard methods.

### 4.3. Immunohistochemistry Analysis

After paraffin removal, histological sections (8–10 μm) were hydrated and incubated for 10 min in TBS buffer (5 mM Tris, 136 mM NaCl, pH 7.4) with 0.01% Triton X-100. Endogenous peroxidase activity was inactivated by treating samples with 3% H_2_O_2_ in methanol for 20 min. After washing the samples in TBS (3 × 5 min), non-specific binding sites were blocked for 40 min with 50 μg of BSA and pure rabbit serum diluted 1:40 in TBS buffer. Sections were incubated with specific primary antibodies (Table 3) diluted 1:100 in TBS overnight at 4 °C in a humid chamber and dark. After washing in TBS (3 × 5 min), sections were incubated with specific peroxidase-conjugated anti-IgG (Sigma) diluted 1:1000 in TBS, for 90 min at room temperature, also in a humid chamber and dark. The samples were washed in TBS (3 × 5 min) before incubation with peroxidase-anti-peroxidase complexes (Sigma) diluted 1:200 in TBS for 1 h at room temperature in a humid chamber and dark. The sections were washed (3 × 5 min) in TBS and incubated for 10 min with 3,3′-diaminobenzidine tablets (SIGMAFAST^TM^, Sigma). Finally, the samples were counterstained with hematoxylin (5 min), dehydrated, and mounted in aqueous mounting medium.

### 4.4. Image Analysis

Histological results were observed using a binocular bright field optical microscope (Nikon Eclipse E400, Madrid, Spain). The microscope had a DS-Fi1 camera, and images were obtained using Toup View 3.7 software. The images obtained were analyzed using FIJI software Image J for semi-quantitative determination of the DAB and hematoxylin signals independently in each image, accordingly with the protocol of Crowe and Yue (2019) [52]. A total of 10 images (*n* = 10) were analyzed for each experimental condition (vehicle or melatonin) and histological staining (β-Tubulin III, β-Amyloid (1-42), Nestin, TBR-2, and NeuroD1) at 400× magnifications. The results are expressed in percentages (with respect to control mice) as the intensity of the DAB signal per total number of nuclei in the cortex and hippocampal dentate gyrus (including the ML, GNL, SGZ, and the hilus). The nuclear volume was done specifically in the GNL and SGZ by counting the number of nuclei in each hematoxylin image at 1000× magnifications. Cells positive for β-Tubulin III were directly counted specifically in the GNL and SGZ of the hippocampus at 1000× magnifications. Frequency of cortical and hilar cells positive for β-Tubulin III with evident neural processes were calculated at 400× magnifications. Length of neural prolongations in cortical and hilar cells positive for β-Tubulin III were measured at 400× magnifications

### 4.5. Statistical Analysis

Statistical analyses were performed using GraphPad Prism 6. Data are presented as the mean values ± S.E.M. calculated from at least three separate experiments. The normality of the data was analyzed using the Kolmogorov–Smirnov test. Mean comparisons were analyzed using Student’s *t*-test to compare means between control mice and melatonin-treated mice. The level of significance was *p* < 0.05. Statistical analysis was always performed in 10 images obtained from each SAMP8 control mice (*n* = 4) and SAMP8 mice treated with melatonin (*n* = 4).

The statistical software package SPSS 15.0 for Windows (SPSS Inc., Chicago, IL, USA) was used for the PCA. The number of components retained was based on eigenvalues (i.e., the amount of the total variance that is explained by each component) of 1 or greater. A varimax rotation was used to obtain a set of independent and best interpretable components and minimize the number of variables that had high saturations in each component. The components were interpreted based on the loadings that related the parameter to the components. Loadings greater than 0.5 were used to identify the variables comprising a component because this cutoff point provides good separation of the components, as previously shown [53]. The Kaiser–Meyer–Olkin measure and Bartlett’s test of sphericity were calculated to evaluate the significance and adequacy of our factor model. Pearson’s correlations among the study variables and their statistical significance (*p*-values) were also calculated.

## 5. Conclusions

The neurodegenerative process that occurs in the hippocampus of SAMP8 mice during accelerated senescence may be a key stimulus to promote a neurogenic response in the hippocampus in order to recover the dead/damaged neurons and avoid major cognitive impairment. However, compromised survival or aberrant migration of neural precursor cells and impairments in neuronal maturation processes seem to promote limited hippocampal neurogenesis in the aged and neurodegenerative brains of 10 months-old SAMP8 mice. This hypothesis was firmly supported by the results obtained in melatonin-treated SAMP8 mice. Chronic treatment with melatonin for 9 months induced a significant decrease in the neurogenerative processes and hippocampal neurogenic response in these mice. However, melatonin seemed to promote cell survival and restoration of the alterations of migration and/or maturation of neural precursor cells observed in the hippocampus of aged control mice, thus promoting a functional adult hippocampal neurogenesis in melatonin-treated mice. Therefore, we corroborate the neuroprotective benefits of chronic treatments with melatonin against the alterations in adult hippocampal neurogenesis induced by neurodegeneration that occur in the brain during aging (see Graphical abstract).

## Figures and Tables

**Figure 1 molecules-27-05543-f001:**
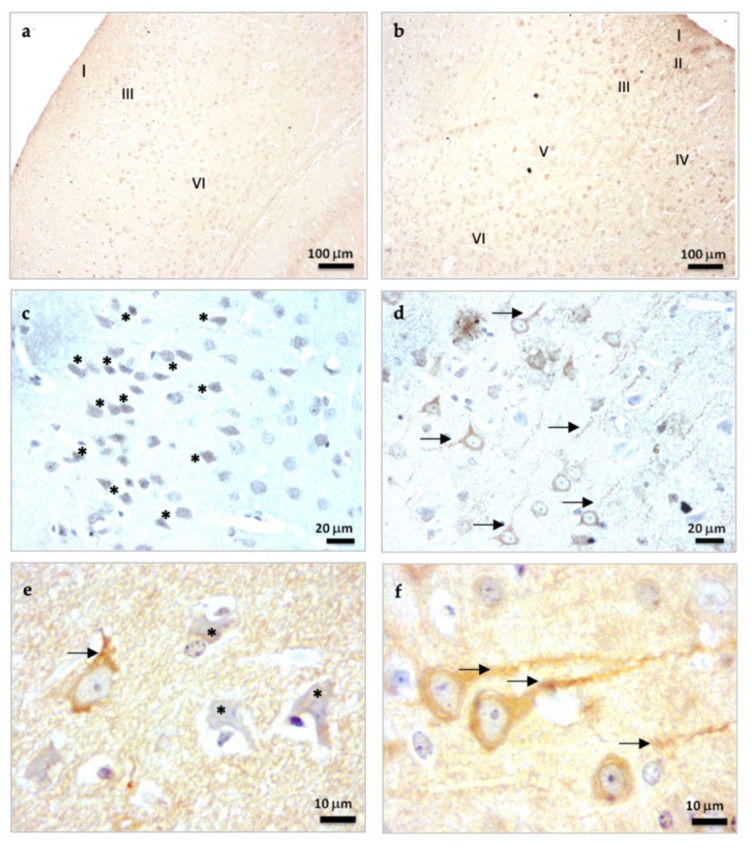
β-Tubulin III immunostaining in the brain cortex of SAMP8 control mice (**a**,**c**,**e**) and SAMP8 mice treated with melatonin (**b**,**d**,**f**). Asterisks show neurons under neurodegeneration (dark neurons). Arrows note neuronal prolongations in cells positive for β-Tubulin III. I, the molecular layer; II, the outer granular layer; III, the outer pyramidal layer; IV; the inner granular layer; V, the inner pyramidal layer; VI, the polymorph layer. (**g**) Bar chart shows quantification of the DAB signal with respect to the total number of nuclei at 400× magnifications (in percentages with respect to control). (**h**) Frequency of cortical cells positive for β-Tubulin III with evident neural processes. (**i**) Length of neural prolongations in cortical cells positive for β-Tubulin III. Data are expressed as means ± SEM. *** *p* < 0.001 vs. control. Statistical analysis was always performed in 10 images obtained from each SAMP8 control mice (*n* = 4) and SAMP8 mice treated with melatonin (*n* = 4).

**Figure 2 molecules-27-05543-f002:**
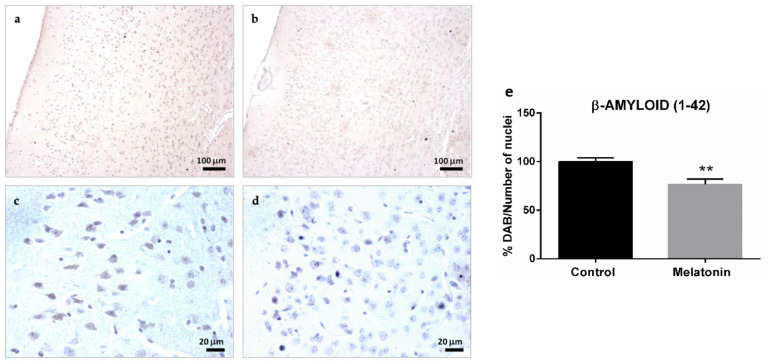
β-Amyloid (1-42) immunostaining in the cortex of SAMP8 control mice (**a**,**c**) and SAMP8 mice treated with melatonin (**b**,**d**). Bar chart (**e**) shows quantification of the DAB signal with respect to the total number of nuclei at 400× magnifications (in percentages with respect to the control). Data are expressed as means ± SEM. ** *p* < 0.01 vs. control. Statistical analysis was always performed in 10 images obtained from each control mice (*n* = 4) and mice treated with melatonin (*n* = 4).

**Figure 3 molecules-27-05543-f003:**
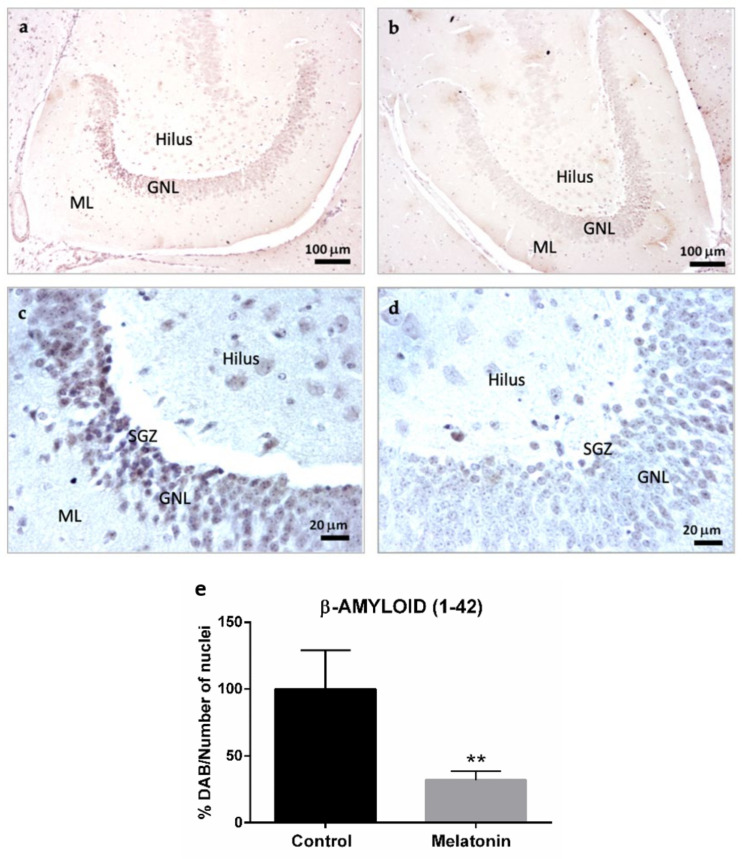
β-Amyloid (1-42) immunostaining in the hippocampal dentate gyrus of SAMP8 control mice (**a**,**c**) and SAMP8 mice treated with melatonin (**b**,**d**). Bar chart (**e**) shows quantification of the DAB signal with respect to the total number of nuclei at 400× magnifications (in percentages with respect to the control). Data are expressed as means ± SEM. ** *p* < 0.01 vs. control. ML, the molecular layer; GNL, the granule neurons layer; SGZ, the subgranular zone. Statistical analysis was always performed in 10 images obtained from each control mice (*n* = 4) and mice treated with melatonin (*n* = 4).

**Figure 4 molecules-27-05543-f004:**
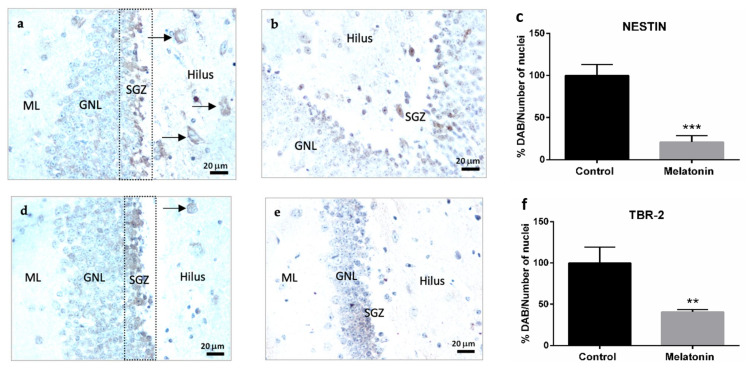
Markers of adult hippocampal neurogenesis in the dentate gyrus of SAMP8 control mice (**a**,**d**,**g**) and SAMP8 mice treated with melatonin (**b**,**e**,**h**). Bar charts (**c**,**f**,**i**) show quantification of the DAB signal with respect to the total number of nuclei at 400× magnifications (in percentages with respect to the control). Data are always expressed as means ± SEM. ** *p* < 0.01; *** *p* < 0.001 vs. control. ML, the molecular layer; GNL, the granule neurons layer; SGZ, the subgranular zone. Statistical analysis was always performed in 10 images obtained from each SAMP8 control mice (*n* = 4) and SAMP8 mice treated with melatonin (*n* = 4).

**Figure 5 molecules-27-05543-f005:**
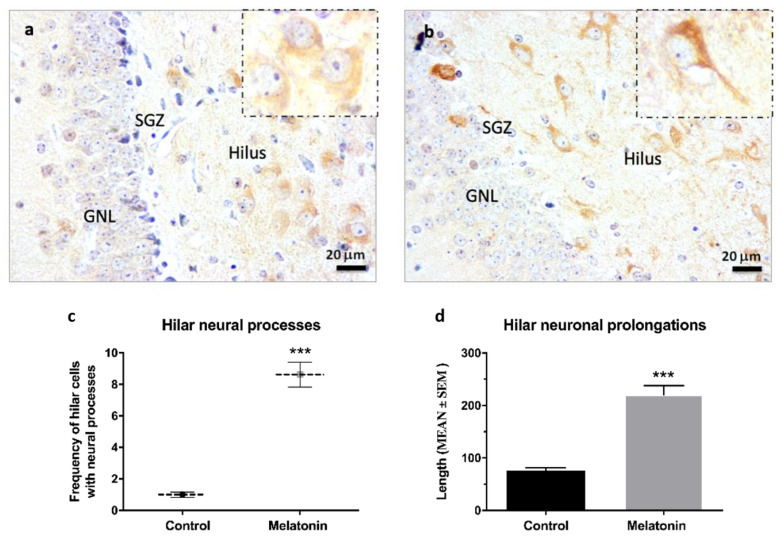
β-Tubulin III immunostaining in the hippocampal dentate gyrus of SAMP8 mice at 400× magnifications. The small box shows details of hilar neurons in control mice (**a**) and melatonin-treated mice (**b**) at 1000× magnifications. GNL, the granule neurons layer; SGZ, the subgranular zone. (**c**) Frequency of hilar cells positive for β-Tubulin III with evident neural processes were calculated at 400× magnifications (**d**) Length of neural prolongations in hilar cells positive for β-Tubulin III were measured at 400× magnifications. Data are expressed as means ± SEM. *** *p* < 0.001 vs. control. Statistical analysis was always performed in 10 images obtained from each SAMP8 control mice (*n* = 4) and SAMP8 mice treated with melatonin (*n* = 4).

**Figure 6 molecules-27-05543-f006:**
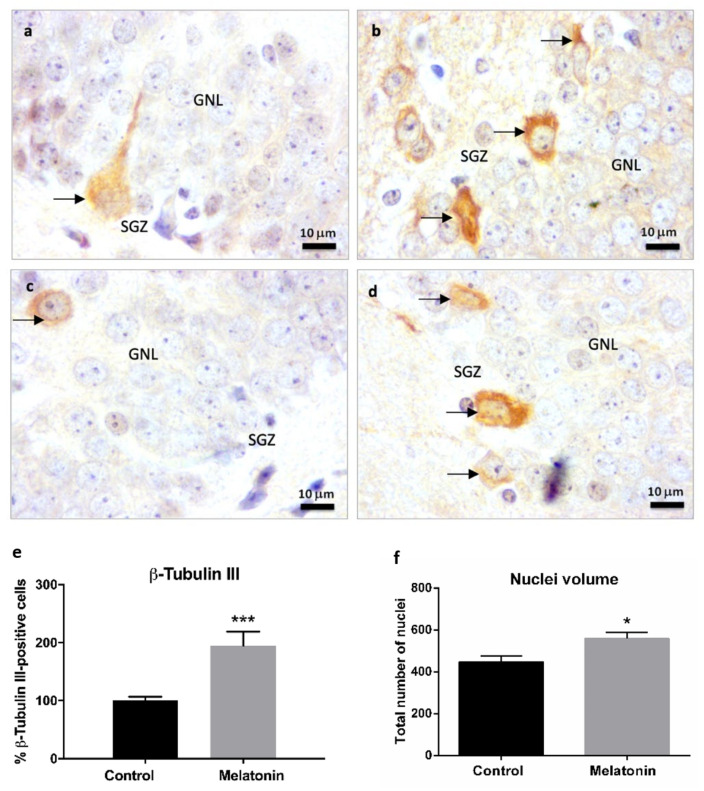
β-Tubulin III immunostaining in the granule neurons layer of the hippocampus of SAMP8 control mice (**a**,**c**) and SAMP8 mice treated with melatonin (**b**,**d**) at 1000× magnifications. The arrows note possible newly created post-mitotic neurons. GNL, the granule neurons layer; SGZ, the subgranular zone. (**e**) Bar chart shows quantification of cells positive for β-Tubulin III in the GNL and SGZ of the hippocampus (in percentages with respect to the control). (**f**) Bar chart shows quantification of the total number of nuclei (nuclei volume) in the GNL and SGZ of the hippocampus. Data are always expressed as means ± SEM. * *p* < 0.05; *** *p* < 0.001 versus control. Statistical analysis was always performed in 10 images obtained from each SAMP8 control mice (*n* = 4) and SAMP8 mice treated with melatonin (*n* = 4).

**Figure 7 molecules-27-05543-f007:**
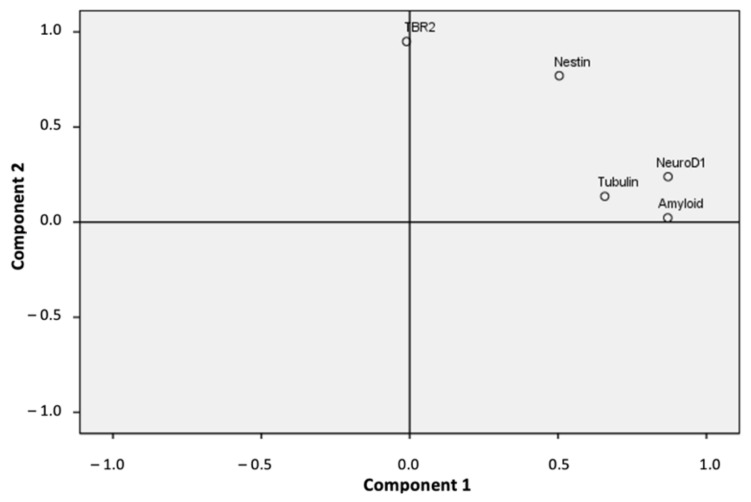
Dimensional representation of the PCA analysis showing the two main components found.

**Table 1 molecules-27-05543-t001:** Descriptive analysis of the rotated component loadings observed after applying a PCA *.

	Component 1	Component 2
β-Amyloid (1-42)	**0.870**	0.023
Nestin	0.504	**0.770**
TBR-2	−0.011	**0.949**
NeuroD1	**0.871**	0.239
β-Tubulin III	**0.657**	0.136

* The Kaiser–Meyer–Olkin measure of sampling adequacy was 0.619; Bartlett’s test of Sphericity showed a significant *p*-value = 0.000.

**Table 2 molecules-27-05543-t002:** Correlations matrix among all the variables included in the PCA.

		β-Amyloid (1-42)	Nestin	TBR-2	NeuroD1	β-Tubulin III
Correlation *	β-Amyloid (1-42)	1	**0.396**	0.098	**0.707**	**0.374**
	Nestin	**0.396**	1	**0.607**	**0.635**	**0.379**
	TBR-2	0.098	**0.607**	1	0.188	0.140
	NeuroD1	**0.707**	**0.635**	0.188	1	**0.424**
	β-Tubulin III	**0.374**	**0.379**	0.140	**0.424**	1
*p*-values *	β-Amyloid (1-42)	-	*0.034*	0.333	*0.000*	*0.043*
	Nestin	*0.043*	-	*0.001*	*0.001*	*0.041*
	TBR-2	0.333	*0.001*	-	0.201	0.268
	NeuroD1	*0.000*	*0.001*	0.201	-	*0.025*
	β-Tubulin III	*0.043*	*0.041*	0.268	*0.025*	-

* Significant Pearson’s correlations are shown in bold; significant *p*-values are shown in italics.

**Table 3 molecules-27-05543-t003:** Primary antibodies used for immunohistochemistry analyses.

Name	Code	Company
Anti-β-Tubulin III (TUBB3)	T2200	Sigma-Aldrich
β-Amyloid (1-42) (D9A3A)	#14974	Cell Signaling
Anti-Nestin	N5413	Sigma-Aldrich
Anti-TBR-2	#ABN1687	Millipore
Anti-NeuroD1	#ABE991	Millipore

## Data Availability

Not applicable.

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
