# Peer review of "Chronic Treatment with Melatonin Improves Hippocampal Neurogenesis in the Aged Brain and Under Neurodegeneration"

_molecules, 2022, doi:10.3390/molecules27175543_

Round 1
Reviewer 1 Report
Cachán-Vega and colleagues have studied the effect of melatonin chronic treatment on neurogenesis in SAMP8 mice model. This group had experience in the effect of melatonin in SAMP8 mice. Here, they showed that administration of melatonin for 9 months reduces the amount of beta-amyloid in the cortex. They also observed a reduction of beta-amyloid in the dentate gyrus. This reduction is accompanied by a decrease of nestin, TBR-2 and NeuroD1 markers and by an increase of beta-tubulin. The results are interesting and clearly presented. However, the manuscript showed some limitations: the low number of “n” per group and the lack of association between the results presented and the effect of melatonin on AD phenotype. Additionally, methods section could be improved. I have the following concerns:
- What is the correlation between the effect of melatonin on hippocampus neurogenesis and the effect of melatonin on cortex presented in figures 1 and 2? Please, this point should be more discussed.
- In discussion section, authors described that SAMP8 mice showed higher expression of Nestin, TBR-2 and NeuroD1. Are the marker levels of SAMP8 mice compared with other model to conclude that higher expression is present? Please clarify this point.
- Authors described that melatonin reduced the levels of beta-amyloid in the cortex and hippocampus. I suggest to include another AD marker (such as phospho-tau) to reinforce this important result. Previously, this group published that melatonin treatment modulates the levels of phospho-tau, but if I well understood, this observation was in the whole brain.
- In Materials section, authors described that 4 newborn SAMP8 mice are collected for the treatment with melatonin or vehicle. Please specifiy whether n=4 is the total number of animals or the number of animals per group.
- In materials section, authors described that quantification is performed in the subgranular zone. However, in some figure legends, authors described that quantification was performed in the subgranular zone and the granular layer. Please clarify this point and specify for each quantification which layer was quantified.
- the chronic administration of melatonin could be result in side effects? This point should be clarify in discussion section. Thanks.
Author Response
Thanks a lot for your relevant comments. See our responses in the pdf submitted.

Reviewer 2 Report
Article
Molecules -1797631
Chronic treatment with melatonin improves hippocampal neurogenesis in the aged brain and under neurodegeneration.
This research presents exciting results, which provide valuable information on the role of chronic treatment with melatonin and their positive impact on the neurodegenerative process and in the recovery of the functionality of the adult brain and aging mice.
Suggests
1. Could you make an approximate calculation to know the melatonin doses received each day?
2. About your tubulin findings:
I consider that if the authors determine the frequency of cells with projections will add additional and elegant support to the results that in animals with melatonin treatment, the tubulin β III is organized in neuronal projections, and the discussion will be helpful for the reader the interpretation of that the findings potential that the sense increased neuronal connectivity.
You could focus on the discussion by highlighting your results and emphasizing new data.
3. Review the wording of section 4.5 of the statistical analysis for a better understanding.
4. The figures must follow the resolution guidelines of the journal. it is necessary to improve these
5. In the material and methods section: add the microscope characteristics used to acquire the images.
Author Response

(The authors gave the same response as above.)

Reviewer 3 Report
In this manuscript, Cachán-Vega et al investigate the impact of chronic administration of melatonin on several markers of neurodegeneration and neurogenesis in SAMP8 mice, an animal model of accelerated senescence.
Major issue
- no control with mice with"normal" aging is provided
- although people do, it is not correct to quantify DAB IHC by measuring the intensity of the signal. reasons can be found, for example, in the following discussion
https://www.researchgate.net/post/IHC_with_DAB_appears_to_be_a_heavily_debated_topic_when_it_comes_to_quantification_If_not_possible_to_quantify_what_inferences_can_be_made_instead
- the PCA analysis at the end of the paper is not clear at all
Minor issue
- Figure 1Aa and b don't really allow appreciating the disorganization of the cortical layers
Author Response

(The authors gave the same response as above.)

Round 2
Reviewer 3 Report
The Authors have acknowledged but not addressed my concerns. The overall readability of the manuscript has improved.
